# Infantile subdural hematoma in Japan: A multicenter, retrospective study by the J-HITs (Japanese head injury of infants and toddlers study) group

**Nobuyuki Akutsu**[1], **Masahiro Nonaka**[2]*, **Ayumi Narisawa**[3], **Mihoko Kato**[4], **Atsuko Harada**[5], **Young-Soo Park**[6]

1 Department of Neurosurgery, Hyogo Prefectural Kobe Children's Hospital, Hyogo, Japan, 2 Department of Neurosurgery, Kansai Medical University, Osaka, Japan, 3 Department of Neurosurgery, Sendai City Hospital, Miyagi, Japan, 4 Department of Neurosurgery, Aichi Children's Health and Medical Center, Aichi, Japan, 5 Department of Pediatric Neurosurgery, Takatsuki General Hospital, Osaka, Japan, 6 Department of Neurosurgery, Nara Medical University, Nara, Japan

* nonakamasa65@gmail.com

**Data Availability Statement:** Data cannot be shared publicly because there is still a possibility of

## Abstract

### Objective

Subdural hematoma in infants or toddlers has often been linked to abuse, but it is not clear how many cases actually occur and how many are suspected of abuse. The purpose of this study was to investigate subdural hematoma in infants and toddlers in Japan.

### Methods

This multicenter, retrospective study reviewed the clinical records of children younger than 4 years with head trauma who were diagnosed with any finding on head computed tomography (CT) and/or magnetic resonance imaging (MRI), such as skull fracture and/or intracranial injury. A total of 452 children were included. The group suspected to have been abused was classified as nonaccidental, and the group considered to have been caused by an accident was classified as accidental. Subdural hematoma and other factors were examined on multivariate analysis to identify which factors increase the risk of nonaccidental injuries.

### Results

Of the 452 patients, 158 were diagnosed with subdural hematoma. Subdural hematoma was the most common finding intracranial finding in head trauma in infants and toddlers. A total of 51 patients were classified into the nonaccidental group, and 107 patients were classified into the accidental group. The age of patients with subdural hematoma showed a bimodal pattern. The mean age of the accidental group with subdural hematoma was significantly older than that in the nonaccidental group (10.2 months vs 5.9 months, respectively. $p < 0.001$). Multivariate analysis showed that patients under 5 months old, retinal hemorrhage, and seizure were significant risk factors for nonaccidental injury (odds ratio (OR) 3.86, $p = 0.0011$; OR 7.63, $p < 0.001$; OR 2.49, $p = 0.03$; respectively). On the other hand,

identifying specific individuals, even though the information is anonymized, since the current study examined the mechanism of injury and multiple clinical findings. Japan's Act on Protection of Personal Information prohibits the release of information containing personal information to the public, including those whose purpose of use is not clear. Data are available from the Department of Neurosurgery, Kansai Medical University at kansaiidainougeka@gmail.com for researchers who meet the criteria for access to confidential data.

**Funding:** The author(s) received no specific funding for this work.

**Competing interests:** Masahiro Nonaka and Young-Soo Park have written statements and appeared in court in child abuse cases both at the request of the prosecutor and the defense. Atsuko Harada has written statements and appeared in court in child abuse cases at the request of the prosecutor. This does not alter our adherence to PLOS ONE policies on sharing data and materials.

the odds ratio for subdural hematoma was 1.96, and no significant difference was observed ($p = 0.34$).

## Conclusions

At least in Japanese children, infantile subdural hematoma was frequently observed not only in nonaccidental but also in accidental injuries. In infants with head trauma, age, the presence of retinal hemorrhage, and the presence of seizures should be considered when determining whether they were abused. Subdural hematoma is also a powerful finding to detect abuse, but care should be taken because, in some ethnic groups, such as the Japanese, there are many accidental cases.

## Introduction

Head trauma is one of the leading causes of pediatric emergency department visits, and the diagnosis of abusive head trauma (AHT) by neurological imaging is difficult in practice. Many studies have suggested that subdural hematoma is considered characteristic of AHT [1–5]. In Japan, the Ministry of Health, Labor and Welfare publishes the "Guide to Responding to Child Abuse" as a standard for child guidance centers to take temporary custody; the 2014 revision states that the case of the infant with subdural hematoma is highly likely to have been abused and should be treated with suspicion. However, the true incidence of subdural hematoma due to head trauma in infants and toddlers, regardless of abuse or accident, is unknown. In this study, the extent of subdural hematoma in cases where imaging studies were performed after head trauma was examined, and the findings and the extent to which abuse was suspected and not suspected are presented.

## Methods

This multicenter, retrospective study reviewed clinical records of children younger than 4 years with head trauma who visited our institutions between January 2014 and August 2020. Patients with some imaging findings such as fracture or intracranial injury were included in the study. Patients without imaging or with only extracranial findings or no obvious findings on imaging were excluded. Two university hospitals, two children's hospitals, and two general hospitals participated in this study, making it possible to obtain data closer to the real world in Japan. A total of 452 children were included in this study. From the medical charts, the sex and age of the child, mechanism of injury, physical and neurological findings, radiological findings, retinal hemorrhage, surgical intervention, notification to child guidance centers, temporary protection by child guidance centers, and criminal cases were extracted. The imaging studies examined in this study included not only computed tomography (CT), but also magnetic resonance imaging (MRI). Imaging findings were confirmed and recorded by a board-certified pediatric neurosurgeon at each institution. The information about the mechanism of injury was primarily determined from the medical history provided by the caregiver to a physician. If necessary, medical evaluation reports by child protection teams at each facility and adequate investigation reports by police and/or child guidance centers were requested. Because many head injuries in infancy are suspected to be caused by abuse, institutions with a child protection team to assess whether the head injury was caused by abuse were included. Patients were classified into the nonaccidental or accidental groups by our definition (Table 1).

**Table 1. Definition of "nonaccidental" and "accidental".**

| Nonaccidental | | | Accidental | | | |
|---|---|---|---|---|---|---|
| Perpetrator confesses to abuse. | Perpetrator did not confess to abuse, but was prosecuted | CPT at each facility determined possible abuse, child guidance center notified, judged as possible abuse, taken into temporary custody | CPT at each facility determined possible abuse, child guidance center notified, judged as accident, not taken into custody | CPT at each facility determined no potential for abuse, and not reported to the child guidance center | Birth injury | Traffic accidents |

Abbreviations: CPT, child protection team.

Our definition includes cases that have been taken into temporary custody by the Child Guidance Center, as well as cases that have been determined to be accidents and have not been notified to the Child Guidance Center. We cannot deny the possibility that some of these cases may be accidents or cases of abuse. For this reason, we also have studied cases of obvious abuse (those who confessed and those who were criminally prosecuted) with cases of traffic accidents, and the results were compared with the results from our classification.

### IRB/ethics committee approval and a statement regarding patient consent

The protocol for this study was approved by the Ethics Committee of Kansai Medical University (No. 2019232). Need for written patient consent was waived by the ethics committee because data were deidentified. Institutional review board approval was obtained from all participants' institutions prior to submitting cases for this study.

### Statistical analysis

Statistical analysis was performed with JMP 14.2.0. Continuous variables were analyzed using the Wilcoxon rank-sum test. Univariate and multivariate analyses were performed to examine the relationship between nonaccidental injury and prognostic factors. Univariate logistic model was used to compare each prognostic factors. Variables were included in a multivariate logistic regression model if their $p$ value in the univariate analysis was statistically significant. The odds ratios (ORs) and 95% confidence intervals (CIs) were calculated.

### Results

Of the 452 patients, 58 were included in the nonaccidental group, and 394 patients were included in the accidental group. The demographics and age distribution of the patients are shown in Table 2 and Fig 1, respectively. The mean age of the nonaccidental group was 5.8 months, and the mean age of the accidental group was 12.1 months ($p < 0.001$). The number of patients in the nonaccidental group was highest at 2 months and 4 months, and in the accidental group at 4 months (Fig 1). Of the 452 patients, 158 were diagnosed with subdural hematoma. Subdural hematoma was the second most common finding on CT from head trauma in infants after skull fracture, and it was more common than epidural hematoma (Table 2). The causes of injury in the accidental group are summarized and described in Table 3.

The mean age was 8.8 months, and the number of patients showed a bimodal peak at 4 months and at 8 months (Fig 2A). Of the patients with subdural hematoma, 51 were classified as the nonaccidental group and 107 as the accidental group. The mean age of patients in the nonaccidental group was 5.9 months, and the mean age of patients in the accidental group was 10.2 months ($p < 0.001$). The number of patients was highest at 2 months and 4 months in the nonaccidental group (Fig 2B) and at 8 months and 10 months in the accidental group (Fig 2C). Retinal hemorrhage was present in 45 of 58 patients in the nonaccidental group and 40 of 394 patients in the accidental group.

**Table 2. Comparison of demographic and clinical presentations.**

| | | Total | Nonaccidental group | Accidental group |
|---|---|---|---|---|
| Total no | | 452 | 58 (12.8%) | 394 (87.2%) |
| Mean age (months) | | 11.3 | 5.8 | 12.1 |
| Male | | 305 | 40 (13.1%) | 265 (83.9%) |
| Female | | 147 | 18 (12.2%) | 129 (87.8%) |
| SDH | total | 158 | 51 (32.3%) | 107 (67.7%) |
| | bilateral SDH | 50 | 25 (50.0%) | 25 (50.0%) |
| | unilateral SDH | 108 | 26 (24.1%) | 82 (75.9%) |
| Skull fracture | | 306 | 18 (5.9%) | 288 (94.1%) |
| EDH | | 100 | 4 (4.0%) | 96 (96.0%) |
| SAH | | 55 | 11 (20.0%) | 44 (80.0%) |
| Contusion | | 33 | 5 (15.2%) | 28 (84.8%) |
| Brain edema | | 35 | 15 (42.9%) | 20 (57.1%) |
| Retinal hemorrhage | yes | 85 | 45 (52.9%) | 40 (47.1%) |
| | no | 154 | 12 (7.8%) | 142 (92.2%) |
| | not examined | 213 | 1 (0.5%) | 212 (99.5%) |
| Seizure | yes | 78 | 34 (43.6%) | 44 (56.4%) |
| | no | 374 | 24 (6.4%) | 350 (93.6%) |
| Surgery | | 85 | 31 (36.5%) | 54 (63.5%) |

Abbreviations: SDH, subdural hematoma; EDH, epidural hematoma; SAH, subarachnoid hemorrhage.

On univariate analysis, age under 5 months (OR 2.83, $p < 0.001$), subdural hematoma (OR 19.54, $p < 0.001$), retinal hemorrhage (OR 13.31, $p < 0.001$), brain edema (OR 6.52, $p < 0.001$), seizure (OR 12.10, $p < 0.001$), and surgical case (OR 7.22, $p < 0.001$) were significantly associated with increased odds of nonaccidental group (Table 4). On the other hand, the odds ratio of the nonaccidental group was significantly lower in cases with skull fracture (OR 0.17, $p < 0.001$) and epidural hematoma (OR 0.23, $p = 0.006$). A univariate analysis was conducted on 22 cases in the apparent abuse group, which included both confessed and criminally prosecuted cases, and 39 cases of traffic trauma, for a total of 61 cases. The results showed that there were significant differences in age under 5 months (OR 8.65, p = 0.0005), subdural

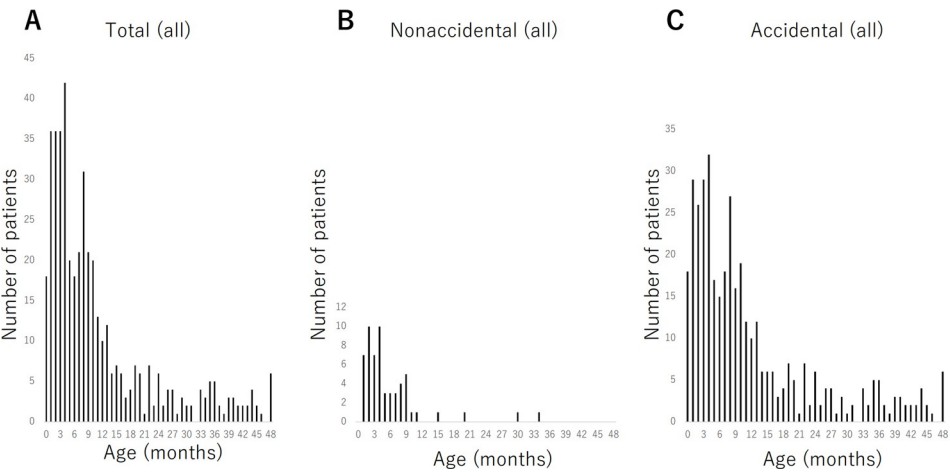

**Fig 1. Age distribution of all patients in this study.** A: Total, B: Nonaccidental, C: Accidental.

**Table 3. List of cause of injuries in accidental groups.**

| Cause of injury | | No.(%) |
|---|---|---|
| Birth injury | | 8(2.0%) |
| Traffic accident | Motor vehicle accident | 27(6.9%) |
| | Bicycle accident | 12(3.0%) |
| Falls from >2m | | 28(7.11%) |
| Falls from ≤2m | Caretakers tumbled while holding the child | 42(10.7%) |
| | Accidentally dropped by caretakers | 89(22.6%) |
| | Self-inflicted tumbles | 38(9.6%) |
| | Falling from couches or bed | 33(8.4%) |
| | Other falls from ≤2m | 91(23.1%) |
| Other injury | | 14(3.6%) |
| Unknown | | 12(3.0%) |
| Total | | 394 |

hematoma (OR 45.71, p < 0.001), retinal hemorrhage (OR 15.30, p = 0.0034), brain edema (OR 4.71, p < 0.017), seizure (OR 18.75, p < 0.001), and surgical case (OR 40.8, p < 0.001) (Table 5). Similarly, the odds ratio of the apparent abuse group was significantly lower in cases with skull fracture (OR 0.10, *p* = 0.0002). Epidural hematoma (OR 0.12, *p* = 0.051) was just barely significant, but the OR was similarly low. These results were similar to the results of our analysis of cases classified as accidental and nonaccidental.

Of the 158 cases of subdural hematoma, 127 (80.4%) underwent fundus examination and were included in the multivariate analysis. On multivariate analysis, age under 5 months, retinal hemorrhage, and seizure were significantly associated with increased odds of nonaccidental injury (OR 3.86, *p* = 0.0011; OR 7.63, *p* < 0.001; OR 2.49, *p* = 0.03; respectively). On the other hand, subdural hematoma was not found to be significant (OR 1.96, *p* = 0.34) (Table 6). Multivariate analysis between the apparent abuse group and the accident group was not performed due to the small number of cases.

Sensitivity, specificity, positive predictive value, and negative predictive value for subdural hematoma, retinal hemorrhage, and brain edema are summarized in Table 7. The sensitivity, specificity, and positive predictive value of subdural hematoma for the diagnosis of nonaccidental injury were 87.9%, 72.8%, and 32.0%, respectively.

## Discussion

Based on the literature, AHT should be part of the differential diagnosis in infants and toddlers with subdural hematoma. In a systematic review, subdural hematoma and retinal hemorrhage were strongly associated with AHT (OR 8.92, 95% CI 6.77–11.74; and OR 27.12, 95% CI 15.70–46.84, respectively) [2]. Therefore, infants and toddlers with subdural hematoma and/or retinal hemorrhage due to short falls should be thoroughly investigated for possible AHT. In the present study, the results of univariate analysis showed an odds ratio of 19.5, with a 95% CI of 8.6–44.4, which is similar to previous reports. However, on multivariate analysis, the odds ratio dropped to 1.96, with a 95% CI of 0.5–7.73, which was not significant. The reason for this difference was thought to be the confounding of subdural hematoma with factors such as age, presence of retinal hemorrhage, and presence of seizures. Furthermore, in the present study, subdural hematoma was found in 87.9% of cases in the nonaccidental group, and child abuse represented 32.2% of all traumatic subdural hematomas. Unlike reports from other countries, we therefore consider that subdural hematoma is frequently associated with, but is not characteristic of AHT in Japan.

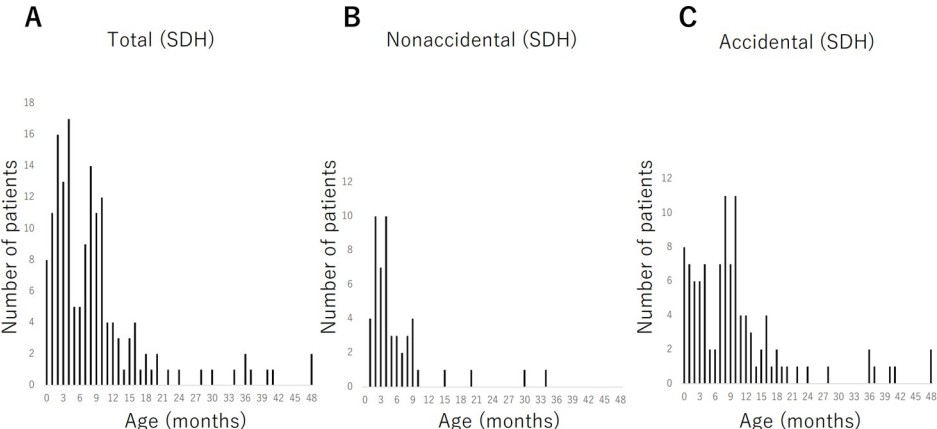

**Fig 2. Age distribution of patients with subdural hematoma.** A: Total, B: Nonaccidental, C: Accidental. There is a peak at 2 to 4 of months of age in the nonaccidental group and another peak at 8 to 10 of months of age in the accidental group.

In Japan, subdural hematoma has long been known to occur due to minor head trauma in infants who are old enough to pull themselves up. This was known as Nakamura's type 1, and many Japanese neurosurgeons were aware of its existence [6, 7]. However, all of these reports have been criticized for not being fully investigated for child abuse [8]. More recently, Amagasa et al. reported that subdural hematoma, retinal hemorrhage, and neurological sequelae due to short falls were not seen after witnessed falls in children younger than 2 years in Japan [5]. However, in their paper, only 14 cases of subdural hemorrhage were found, and it is difficult to assume that most subdural hematomas in Japan are caused by abuse. In recent years, there have been similar reports of infantile acute subdural hematoma due to accidental minor head trauma from other countries [9–12]. Hogberg et al. reported that 104 of 306 (34.0%) infants with subdural hematoma younger than 1 year in Sweden were caused by accidental falls [13].

The peak incidence of AHT was reported to be 2 to 4 months of age in studies from North America [14–16]. Nuno et al. explained that the cause of the peak was that frustrated parents often use shaking as a means of quieting an infant, and reports of excessive crying at

**Table 4. Univariate analysis (nonaccidental vs accidental).**

|  | n | Odds ratio | p value | Lower 95% CI | Upper 95% CI |
|---|---|---|---|---|---|
| Age under 5 months | 452 | 2.84 | **0.0004***| 1.60 | 5.03 |
| Male | 452 | 1.17 | 0.60 | 0.65 | 2.12 |
| Subdural hematoma | 452 | 19.54 | < **.0001***| 8.60 | 44.40 |
| Retinal bleeding | 239 | 13.31 | < **.0001***| 6.43 | 27.54 |
| Brain edema | 452 | 6.52 | < **.0001***| 3.11 | 13.60 |
| Skull fracture | 452 | 0.17 | < **.0001***| 0.09 | 0.30 |
| Contusion | 452 | 1.236 | 0.68 | 0.46 | 3.33 |
| Epidural hematoma | 452 | 0.23 | **0.006***| 0.08 | 0.65 |
| Subarachnoid hemorrhage | 452 | 1.86 | 0.09 | 0.90 | 3.85 |
| Seizure | 452 | 12.10 | < **.0001***| 6.56 | 22.33 |
| Surgery | 452 | 7.23 | < **.0001***| 4.01 | 13.05 |

*Significant Abbreviations: CI, confidence interval.

**Table 5. Univariate analysis of confessed or accused abused cases vs traffic accidental cases.**

|  | n | odds ratio | *p* value | lower 95% CI | upper 95% CI |
|---|---|---|---|---|---|
| Age under 5 months old | 61 | 8.65 | **0.0005**\* | 2.56 | 29.22 |
| Male | 61 | 2.04 | 0.2 | 0.68 | 6.09 |
| Subdural hematoma | 61 | 45.71 | **< .0001**\* | 8.62 | 242.31 |
| Retinal bleeding | 33 | 15.30 | **0.0034**\* | 2.46 | 95.19 |
| Brain edema | 61 | 4.71 | **0.017**\* | 1.33 | 16.70 |
| Skull fracture | 61 | 0.10 | **0.0002**\* | 0.029 | 0.33 |
| Contusion | 61 | 0.87 | 0.85 | 0.19 | 3.88 |
| Epidural hematoma | 61 | 0.12 | 0.051 | 0.014 | 1.01 |
| Subarachnoid hemorrhage | 61 | 0.74 | 0.65 | 0.20 | 2.76 |
| Seizure | 61 | 18.75 | **< .0001**\* | 4.77 | 73.74 |
| Surgery | 61 | 40.8 | **< .0001**\* | 8.72 | 190.92 |

\*Significant Abbreviations: CI, confidence interval.

approximately 2 months of age coincide with documented cases of AHT [15]. The present study also shows that cases of head trauma less than 5 months old are more likely to have been abused, which is consistent with previous reports. On the other hand, Fujiwara et al. reported that patients with AHT younger than 2 years in Japan had 2 peaks of age, at 2 to 4 months and at 7 to 9 months [3]. Similarly, there are some reports from Japan that the age distribution of the AHT group had two peaks of age [4, 17]. Ganesh et al. also questioned why the Japanese reports of subdural hematoma with retinal hemorrhage after minor trauma were so very different from the bulk of the world's literature [18]. However, it was unclear why the data from Japan showed such a bimodal pattern. Fujiwara et al. suggested that the first peak was due to the association between shaken infant syndrome and the peak of crying, and the cause of the second peak, which was not found in Western countries, was the easy availability and taking of head CT in Japan [3]. Amagasa et al. suggested that intense crying, such as sleep-related night-time crying, which peaks at 6 to 8 months in Japanese infants, and sleep sharing could be possible factors for abuse in older infants in Japan [4]. However, there is no evidence to support their suggestions in these papers. In the present study, 158 cases of subdural hematoma were carefully reviewed by the child protection team of each hospital, and cases suspected of abuse were reported to the child guidance center for full consideration of possible abuse. As a result, 107 cases were determined to be accidental, whereas 51 cases were determined to be nonaccidental. Furthermore, the peak age of the nonaccidental cases was 4 months, which is similar to

**Table 6. Multivariate analysis of nonaccidental vs accidental.**

|  | n | odds ratio | p value (Prob>ChiSq) | lower 95% CI | upper 95% CI |
|---|---|---|---|---|---|
| Under 5 months old | 452 | 3.86 | **0.0011**\* | 1.71 | 8.71 |
| Subdural hematoma | 452 | 1.96 | 0.34 | 0.5 | 7.73 |
| Retinal bleeding | 239 | 7.63 | **< .0001**\* | 2.76 | 21.2 |
| Brain edema | 452 | 1.29 | 0.61 | 0.48 | 3.52 |
| Skull fracture | 452 | 1.03 | 0.96 | 0.36 | 2.98 |
| Epidural hematoma | 452 | 1.66 | 0.46 | 0.43 | 6.43 |
| Seizure | 452 | 2.49 | **0.03**\* | 1.1 | 5.69 |
| Surgery | 452 | 1.26 | 0.6 | 0.53 | 3.02 |

\*Significant Abbreviations: CI, confidence interval.

**Table 7. Sensitivity, specificity, positive predictive value, and negative predictive value.**

| Example | | | |
|---|---|---|---|
| | **Nonaccidental** | **Accidental** | |
| | A | B | positive predictive value |
| | C | D | negative predictive value |
| | sensitivity | specificity | |
| Subdural hematoma | | | |
| | Nonaccidental | Accidental | |
| yes | 51 | 107 | 0.32 |
| no | 7 | 287 | 0.98 |
| | 0.88 | 0.73 | |
| Retinal hemorrhage | | | |
| | Nonaccidental | Accidental | |
| yes | 45 | 40 | 0.53 |
| no | 12 | 142 | 0.92 |
| | 0.79 | 0.78 | |
| Brain edema | | | |
| | Nonaccidental | Accidental | |
| yes | 15 | 20 | 0.43 |
| no | 43 | 374 | 0.90 |
| | 0.26 | 0.95 | |
| Age under 5 months | | | |
| | Nonaccidental | Accidental | |
| yes | 37 | 151 | 0.20 |
| no | 21 | 243 | 0.92 |
| | 0.64 | 0.62 | |
| Seizure | | | |
| | Nonaccidental | Accidental | |
| yes | 34 | 44 | 0.44 |
| no | 24 | 350 | 0.94 |
| | 0.59 | 0.89 | |

the peak age of abused children in other countries, whereas the peak age of the accidental cases was 8 to 10 months. The second peak in the incidence of subdural hematoma in this study indicates that a large number of accidental head injuries may be included. Furthermore, the previous reports from Japan may have misclassified patients as non-accidental in the older age peak.

Retinal hemorrhage that is bilateral, severe, and includes the posterior pole and peripheral hemorrhage has been reported as characteristic of AHT [19, 20]. In the present study, retinal hemorrhage was also significantly associated with increased odds of AHT. However, an eye examination was not always done within 48 hours of admission in the present study, and subtle retinal hemorrhage might disappear [21]. On the other hand, Scheller reported ten cases of retinal hemorrhage with no evidence of brain injury and suggested that clinicians should reassess the importance of retinal hemorrhage in the setting of suspected AHT [22]. Mechanisms that cause retinal hemorrhage other than vitreoretinal traction, such as abrupt increases in intracranial pressure, have also been suggested [23]. Needless to say, the presence of retinal hemorrhage does not prove AHT, but it is one of the important factors suggesting AHT.

Seizure has also been reported to be associated with AHT. (2) Ichord et al. confirmed that the higher rate of hypoxic-ischemic injury on diffusion-weighted MRI in AHT than in

accidental head trauma is likely multifactorial, involving respiratory insufficiency, seizure, and intracranial space occupying lesions requiring neurosurgical intervention [24]. The higher incidence of seizure in AHT compared to accidental head trauma may also be related to the frequency and the pattern of hypoxic-ischemic injury lesions. Seizure in the acutely injured brain may exacerbate injury directly through excitotoxic mechanisms or indirectly by exacerbating respiratory insufficiency [25]. These studies suggest that management of infants and toddlers with head injury, and particularly from AHT, should include meticulous attention to optimizing ventilation, oxygenation, perfusion, and the diagnosis and treatment of seizure.

The current study has several limitations. First, it had a retrospective design, which can result in misclassification because missing values occur frequently. Although children with intracranial injury were included in the sample, the threshold to suspect intracranial injury and order head CT and/or MRI was not clearly defined a priori. In addition, fundoscopy and skeletal surveys were not completed in some cases. Second, the classification of a child as non-accidental or accidental depends on the judgment of the child guidance center and the CPT of each hospital, which is often based on medical findings. However, there may be a problem with the accuracy of the past reports on which the judgment is based, and as a result, there is a possibility that the judgment may be affected. Third, the nonaccidental group is overrepresented because it includes all cases taken into temporary custody by the child guidance center. Further research should be conducted regarding the differences in the mechanisms of accidental and nonaccidental head injuries in infants and toddlers.

## Conclusions

In the present study in Japan, unlike in other countries, more than half of the infantile subdural hematomas were determined to be accidental. This suggests that the likelihood of accidental subdural hematoma in infants may vary by ethnicity. On the other hand, cases younger than 5 months, cases with retinal hemorrhage, and cases with seizure were found to more likely have been abused, as in other countries. The diagnosis of abuse should not be simply judged by applying the standards of one particular region to other regions, but it is necessary to consider the possibility that there are cultural and racial differences in each region.

## Author Contributions

**Conceptualization:** Masahiro Nonaka, Young-Soo Park.

**Data curation:** Masahiro Nonaka.

**Formal analysis:** Masahiro Nonaka.

**Investigation:** Masahiro Nonaka.

**Methodology:** Masahiro Nonaka.

**Project administration:** Masahiro Nonaka.

**Resources:** Nobuyuki Akutsu, Masahiro Nonaka, Ayumi Narisawa, Mihoko Kato, Atsuko Harada, Young-Soo Park.

**Supervision:** Masahiro Nonaka, Young-Soo Park.

**Validation:** Masahiro Nonaka.

**Visualization:** Masahiro Nonaka.

**Writing – original draft:** Nobuyuki Akutsu.

**Writing – review & editing:** Nobuyuki Akutsu, Masahiro Nonaka, Ayumi Narisawa, Mihoko Kato, Atsuko Harada, Young-Soo Park.

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
