## [Decision Letter · Decision Letter 0]

30 Dec 2021

PONE-D-21-35151Infantile subdural hematoma in Japan: A multicenter, retrospective study by the J-HITs (Japanese Head injury of Infants and Toddlers study) groupPLOS ONE

Dear Dr. Nonaka,

Thank you for submitting your manuscript to PLOS ONE. After careful consideration, we feel that it has merit but does not fully meet PLOS ONE’s publication criteria as it currently stands. Therefore, we invite you to submit a revised version of the manuscript that addresses the points raised during the review process.

We look forward to receiving your revised manuscript.

Kind regards,

Alfio Spina, M.D.

Academic Editor

PLOS ONE

Journal Requirements:

3. You indicated that you had ethical approval for your study. In your Methods section, please ensure you have also stated whether you obtained consent from parents or guardians of the minors included in the study or whether the research ethics committee or IRB specifically waived the need for their consent.

Additional Editor Comments (if provided):

The authors of this paper seek to expound the association of subdural hematoma and non-accidental trauma in infants in toddlers in Japan. The paper gives an interesting look into how Japan differers from the western world in accidental and non-accidental trauma. It also solidifies previously presented ideas that retinal hemorrhages, young age, seizures coupled with subdural hematoma have a high predictive value for non-accidental trauma. Overall, a very interesting paper but could use some added information.

1. It would be interesting to have a section that talks or chart that shows they types of accidental trauma they have, ie. falling out of bed or off a changing table, to see if it differs from other countries. This information would be especially nice to know on the 107 patients with SDH that were determined to be accidental traumas.

2. When you discuss the two peaks of age that has been found previously in Japanese studies but that was not seen in yours, are you attempting to say that maybe the previous studies had patients that were misclassified as non-accidental in the older age peak? This currently isn't clear in the paper

3. Lastly, what is the take away of this article for readers outside of Japan? How does it contribute or educate the global reader? These should be addressed in the conclusion.

Academic Editor: Reviewers found potential in your article. Please revise it according to the attached comments.

Reviewers' comments:

Reviewer's Responses to Questions

5. Review Comments to the Author

Reviewer #1: The authors present a large series of infants and toddlers with evaluation of head trauma to assess an association between subdural hemorrhage and abusive head trauma when accounting for other clinical signs and age. They find a low positive-predictive value, which reflects interactions between prevalence and test/associations.

51 / 452 is a strikingly high rate of non-accidental trauma, but this may be clarified by the selective denominator involving children with abnormal imaging rather than all patients presenting following trauma.

The categorization of non-accidental and accidental is at the authors’ discretion. There may be legal and cultural distinctions in the CPT decisions on taking a child into temporary custody, which is, as illustrated on Table 1, the critical dividing line. When reporting neurosurgical outcomes such as mRS after thrombectomy, the dichotomization has been used to improve statistical power, but the individual categories are also reported for further reader understanding. This may be an appropriate application of that concept as well. Alternatively, the authors may be able to conduct the univariate and multivariate analysis excluding the patients ‘taken into temporary custody’ to demonstrate that the findings are similar and robust, essentially independent of this borderline definition.

Is there a standardized criteria for CPT taking a child into temporary custody? Such as a local or national law or an institutional standard by the overseeing CPT department? If so, a short description of this may be appropriate.

Are there children taken into permanent custody but for which a perpetrator is not prosecuted?

Table 2 would typically have % listed as a total of the column (i.e. male 40 (69%))

In Table 3, if the significant variables had a p-value formatted with bold, it may improve reader focus.

Ultimately, one main critique of this analysis is that the definition of non-accidental and accidental as the gold-standard truth for measuring outcomes. Here the outcome is measured by what clinicians and social workers deem to be suspected AHT. However, these decisions are made based on the constellation of multitrauma and HPI. Therefore, there is a concern about circular reasoning. The study aims to determine how strongly subdurals are associated with AHT. The definition of AHT is clinician and social work judgement of whether AHT exists. This judgement is based on expert opinion with limited data and no ability to confirm a historical event, but the existing opinion includes subdural hemorrhage and retinal hemorrhage as components of a suspicious trauma pattern. The clinicians and social workers therefore use SDH as one feature of their judgement, by which the study outcomes are measured. However, it remains unclear to the reviewer how to extricate the field from this circle – just that it must be acknowledged in the discussion.

Reviewer #2: I appreciate the author presenting this research article article. My comments as as follows

1. The major drawbacks of current stduy are the retrospective design and the nonaccidental group is overrepresented which made the applications of the results to the clinical unpredictable.

2. Table 5: the title is not clear. Sensitivity, specificity, positive predictive value, and negative predictive value ?

Table 5: How about the sensitivity, specificity, positive predictive value, and negative predictive value of seizure and

age under 5 months for diagnosis of AHT

Table 4: why epidural hematoma ans surgery were not evaluated in mutivariate analysis?

Table 2: the tilte is too simple : Overall results?

Fig.1: please clarify what do the x-axis and y-axis represnt?

Reviewer #3: The authors of this paper seek to expound the association of subdural hematoma and non-accidental trauma in infants in toddlers in Japan. The paper gives an interesting look into how Japan differers from the western world in accidental and non-accidental trauma. It also solidifies previously presented ideas that retinal hemorrhages, young age, seizures coupled with subdural hematoma have a high predictive value for non-accidental trauma. Overall, a very interesting paper but could use some added information.

1. It would be interesting to have a section that talks or chart that shows they types of accidental trauma they have, ie. falling out of bed or off a changing table, to see if it differs from other countries. This information would be especially nice to know on the 107 patients with SDH that were determined to be accidental traumas.

2. When you discuss the two peaks of age that has been found previously in Japanese studies but that was not seen in yours, are you attempting to say that maybe the previous studies had patients that were misclassified as non-accidental in the older age peak? This currently isn't clear in the paper

3. Lastly, what is the take away of this article for readers outside of Japan? How does it contribute or educate the global reader? These should be addressed in the conclusion.

6. PLOS authors have the option to publish the peer review history of their article (what does this mean?). If published, this will include your full peer review and any attached files.

Reviewer #1: **Yes: **Pokmeng See

Reviewer #2: No

Reviewer #3: No

---

## [Author Response · Author response to Decision Letter 0]

26 Jan 2022

Reviewer #1: 

Comment #1

The authors present a large series of infants and toddlers with evaluation of head trauma to assess an association between subdural hemorrhage and abusive head trauma when accounting for other clinical signs and age. They find a low positive-predictive value, which reflects interactions between prevalence and test/associations.

51 / 452 is a strikingly high rate of non-accidental trauma, but this may be clarified by the selective denominator involving children with abnormal imaging rather than all patients presenting following trauma.

The categorization of non-accidental and accidental is at the authors’ discretion. There may be legal and cultural distinctions in the CPT decisions on taking a child into temporary custody, which is, as illustrated on Table 1, the critical dividing line. When reporting neurosurgical outcomes such as mRS after thrombectomy, the dichotomization has been used to improve statistical power, but the individual categories are also reported for further reader understanding. This may be an appropriate application of that concept as well. Alternatively, the authors may be able to conduct the univariate and multivariate analysis excluding the patients ‘taken into temporary custody’ to demonstrate that the findings are similar and robust, essentially independent of this borderline definition.

Author’s Response: This suggestion was very helpful to improve the quality of this paper. Thank you very much. In accordance with the reviewer's suggestion, univariate analysis was performed, comparing traffic head injuries (car and bicycle accidents) and cases of apparent abuse (confessed or criminally prosecuted cases). The total number of cases was 61, which was clearly insufficient to perform multivariate analysis, so the multivariate analysis was not performed. The following sentences and table were added.

Change to Text: Lines 103-109 (Revised Manuscript with Track Changes)

Our definition includes cases that have been taken into temporary custody by the Child Guidance Center, as well as cases that have been determined to be accidents and have not been notified to the Child Guidance Center. We cannot deny the possibility that some of these cases may be accidents or cases of abuse. For this reason, we also have studied cases of obvious abuse (those who confessed and those who were criminally prosecuted) with cases of traffic accidents, and the results were compared with the results from our classification.

Change to Text: Lines 162-179 (Revised Manuscript with Track Changes)

On the other hand, the odds ratio of the nonaccidental group was significantly lower in cases with skull fracture (OR 0.17, p < 0.001) and epidural hematoma (OR 0.23, p = 0.006). A univariate analysis was conducted on 22 cases in the apparent abuse group, which included both confessed and criminally prosecuted cases, and 39 cases of traffic trauma, for a total of 61 cases. The results showed that there were significant differences in age under 5 months (OR 8.65, p =0.0005), subdural hematoma (OR 45.71, p < 0.001), retinal hemorrhage (OR 15.30, p =0.0034), brain edema (OR 4.71, p < 0.017), seizure (OR 18.75, p < 0.001), and surgical case (OR 40.8, p < 0.001) (Table 4). Similarly, the odds ratio of the apparent abuse group was significantly lower in cases with skull fracture (OR 0.10 , p =0.0002). Epidural hematoma (OR 0.12, p =0.051) was just barely significant, but the OR was similarly low. These results were similar to the results of our analysis of cases classified as accidental and nonaccidental.

New table has been added (Table 5)

Comment #2

Is there a standardized criteria for CPT taking a child into temporary custody? Such as a local or national law or an institutional standard by the overseeing CPT department? If so, a short description of this may be appropriate.

Author’s Response: In Japan, there is no law stipulating the standards for temporary protection by child guidance centers, but our Ministry of Health, Labor and Welfare's Child Abuse Response Guide, revised in 2014, states that cases of infants with subdural hematomas are highly likely to have been abused and should be treated with suspicion. The following sentences were added.

Change to Text: Lines 65-68 (Revised Manuscript with Track Changes)

In Japan, the Ministry of Health, Labor and Welfare publishes the "Guide to Responding to Child Abuse" as a standard for child guidance centers to take temporary custody; the 2014 revision states that the case of the infant with subdural hematoma is highly likely to have been abused and should be treated with suspicion.

Comment #3

Are there children taken into permanent custody but for which a perpetrator is not prosecuted?

Author’s Response: There are some children who are taken into permanent custody, even if the perpetrator is not prosecuted.

Comment #4

Table 2 would typically have % listed as a total of the column (i.e. male 40 (69%))

Author’s Response: The Table has been revised according to the reviewers' suggestions.

Change to Table: % has been listed as a total of the column of Table 2

Comment #5

In Table 3, if the significant variables had a p-value formatted with bold, it may improve reader focus.

Author’s Response: The Table has been revised according to the reviewers' suggestions.

Change to Table: In Table 4 (Formerly Table 3), significant p-value have been formatted with bold.

Ultimately, one main critique of this analysis is that the definition of non-accidental and accidental as the gold-standard truth for measuring outcomes. Here the outcome is measured by what clinicians and social workers deem to be suspected AHT. However, these decisions are made based on the constellation of multitrauma and HPI. Therefore, there is a concern about circular reasoning. The study aims to determine how strongly subdurals are associated with AHT. The definition of AHT is clinician and social work judgement of whether AHT exists. This judgement is based on expert opinion with limited data and no ability to confirm a historical event, but the existing opinion includes subdural hemorrhage and retinal hemorrhage as components of a suspicious trauma pattern. The clinicians and social workers therefore use SDH as one feature of their judgement, by which the study outcomes are measured. However, it remains unclear to the reviewer how to extricate the field from this circle – just that it must be acknowledged in the discussion.

Author’s Response: The possibility that judgments of abuse may fall into the circular argument is described in the discussion

Change to Text: Lines 304-308 (Revised Manuscript with Track Changes)

Second, the classification of a child as nonaccidental or accidental depends on the judgment of the child guidance center and the CPT of each hospital, which is often based on medical findings. However, there may be a problem with the accuracy of the past reports on which the judgment is based, and as a result, there is a possibility that the judgment may be affected.

Reviewer #2: 

Comment #1

1. The major drawbacks of current study are the retrospective design and the nonaccidental group is overrepresented which made the applications of the results to the clinical unpredictable.

Author’s Response: We believe that the limitation of this paper is the possibility of overrepresented nonaccidental group due to the boundaries we have set. So, in order to check the accuracy of this boundary between nonaccidental and accidental, we compared the results of univariate analysis of apparent abuse and traffic accidents, as described in the response to Reviewer #1, and checked whether the boundary we set between nonaccidental and accidental is reasonable. In both cases, the items that showed a significant difference were almost identical, and we consider that the boundary we divided was reasonable.

Change to Text: Please see response to Reviewer #1’s comment #1

Comment #2

2. Table 5: the title is not clear. Sensitivity, specificity, positive predictive value, and negative predictive value?

Comment #3

Table 5: How about the sensitivity, specificity, positive predictive value, and negative predictive value of seizure and

age under 5 months for diagnosis of AHT

Author’s Response: The title of the Table has been revised according to the reviewers' suggestions. And we have added a table of sensitivity, specificity, positive and negative predictive values for seizure and age under 5 months.

Change to Text: Table 7 (formerly Table 5) has been modified.

Table 4: why epidural hematoma ans surgery were not evaluated in mutivariate analysis?

Author’s Response: Initially, we reduced the number of items to be analyzed in order to improve the accuracy of the multivariate analysis, but as pointed out by the reviewers, we included all items that were significantly different in the univariate analysis. The numbers changed slightly, but the items that showed significant differences remained the same.

Change to Table and Text: Table 6 (formerly Table 4) has been modified. 

Lines 44-52 (Revised Manuscript with Track Changes)

(odds ratio (OR) 3.86, p = 0.0011; OR 7.63, p < 0.001; OR 2.49, p = 0.03; respectively). On the other hand, the odds ratio for subdural hematoma was 1.96, and no significant difference was observed (p = 0.34).

Table 2: the tilte is too simple : Overall results?

Author’s Response: The title of the Table has been revised according to the reviewers' suggestions.

Change to Table: The title of the Table 3 (formerly Table 2) has been modified.

Fig.1: please clarify what do the x-axis and y-axis represnt?

Author’s Response: Thank you for pointing this out. We have filled in what the X and Y axes indicate.

Change to Figure:　In Figure 1 and 2, we have filled in what the X and Y axes indicate.

Reviewer #3: The authors of this paper seek to expound the association of subdural hematoma and non-accidental trauma in infants in toddlers in Japan. The paper gives an interesting look into how Japan differers from the western world in accidental and non-accidental trauma. It also solidifies previously presented ideas that retinal hemorrhages, young age, seizures coupled with subdural hematoma have a high predictive value for non-accidental trauma. Overall, a very interesting paper but could use some added information.

1. It would be interesting to have a section that talks or chart that shows they types of accidental trauma they have, ie. falling out of bed or off a changing table, to see if it differs from other countries. This information would be especially nice to know on the 107 patients with SDH that were determined to be accidental traumas.

Author’s Response: Thank you for your interest in the cause of the injury.We have included in Table 3 the causes of injury in the accident group and the number of patients by cause of injury. 

The reviewer would also like to know what is the cause of the accidental subdural hematoma injury. However, we believe that a detailed study between the cause of injury and the subdural hematoma is needed. In addition, if the cause of injury of subdural hematoma is discussed in this paper, the paper would be too long. Therefore, we are preparing another paper that examines the relationship between the cause of injury and subdural hematoma. Please understand that we do not describe the causes and number of subdural hematoma injuries in this manuscript to prevent the current paper from being dual publication with the other paper that we are working on. We will show a table with data on subdural hematoma in addition to the entire accident group for the reviewer's reference only.

Cause of injury All Subdural Hematoma

Birth injury 8(2.0%) 6(5.6%)

Traffic accident Motor vehicle accident 27(6.9%) 6(5.6%)

 Bicycle accident 12(3.0%) 1(0.9%)

Falls from >2m 28(7.11%) 4(3.7%)

Falls from ≤2m Caretakers tumbled while holding the child 42(10.7%) 9(8.4%)

 Accidentally dropped by caretakers 89(22.6%) 10(9.3%)

 Self-inflicted tumbles 38(9.6%) 24(22.4%)

 Falling from couches or bed 33(8.4%) 18(16.8%)

 Other falls from ≤2m 91(23.1%) 16(14.9%)

Other injury 14(3.6%) 6(5.6%)

Unknown 12(3.0%) 7(6.5%)

Total 394 107

Change to Text: Lines 101-102 (Unmarked); Lines 112-113 (Revised Manuscript with Track Changes)

The causes of injury in the accidental group are summarized and described in Table 3.

In the Table 3 (new Table), list of cause of injuries in accidental group have been presented.

2. When you discuss the two peaks of age that has been found previously in Japanese studies but that was not seen in yours, are you attempting to say that maybe the previous studies had patients that were misclassified as non-accidental in the older age peak? This currently isn't clear in the paper

Author’s Response: We tried not to be direct, but it may have been confusing and inappropriate. We have clearly stated that the previous studies may have misclassified patients as non-accidental in the older age peak as indicated.

Change to Text: Lines 275-276 (Revised Manuscript with Track Changes)

Furthermore, the previous reports from Japan may have misclassified patients as non-accidental in the older age peak

3. Lastly, what is the take away of this article for readers outside of Japan? How does it contribute or educate the global reader? These should be addressed in the conclusion.

Author’s Response: Our results show that there are regional differences in subdural hematoma in infants. The reason for this is unknown, but it is necessary to consider the possibility that there are cultural and racial differences. Therefore, it is necessary to discuss the diagnosis of abuse based on the actual situation in each region.

Change to Text: Lines 318-321 (Revised Manuscript with Track Changes)

The diagnosis of abuse should not be judged by applying the standards of one particular region to other regions, but it is necessary to consider the possibility that there are cultural and racial differences in each region.

From editorial office:

3. You indicated that you had ethical approval for your study. In your Methods section, please ensure you have also stated whether you obtained consent from parents or guardians of the minors included in the study or whether the research ethics committee or IRB specifically waived the need for their consent.

Change to Text: Lines 110-115 (Revised Manuscript with Track Changes)

IRB/ethics committee approval and a statement regarding patient consent

The protocol for this study was approved by the Ethics Committee of Kansai Medical University (No. 2019232). Need for written patient consent was waived by the ethics committee because data were deidentified. Institutional review board approval was obtained from all participants’ institutions prior to submitting cases for this study.

---

## [Editor Report · Decision Letter 1]

10 Feb 2022

Infantile subdural hematoma in Japan: A multicenter, retrospective study by the J-HITs (Japanese Head injury of Infants and Toddlers study) group

PONE-D-21-35151R1

Dear Dr. Nonaka,

We’re pleased to inform you that your manuscript has been judged scientifically suitable for publication and will be formally accepted for publication once it meets all outstanding technical requirements.

Kind regards,

Alfio Spina, M.D.

Academic Editor

PLOS ONE
---

## [Editor Report · Acceptance letter]

16 Feb 2022

PONE-D-21-35151R1 

Infantile subdural hematoma in Japan: A multicenter, retrospective study by the J-HITs (Japanese Head injury of Infants and Toddlers study) group 

Dear Dr. Nonaka:

I'm pleased to inform you that your manuscript has been deemed suitable for publication in PLOS ONE. Congratulations! Your manuscript is now with our production department. 

Kind regards, 

on behalf of

Dr. Alfio Spina 

Academic Editor

PLOS ONE